# Analysis of computational codon usage models and their association with translationally slow codons

**Gabriel Wright**[1]*, **Anabel Rodriguez**[2], **Jun Li**[3], **Patricia L. Clark**[2], **Tijana Milenković**[1],
**Scott J. Emrich**[4]

**1** Department of Computer Science & Engineering, University of Notre Dame, Notre Dame, IN, United States of America, **2** Department of Chemistry & Biochemistry, University of Notre Dame, Notre Dame, IN, United States of America, **3** Department of Applied and Computational Mathematics and Statistics, University of Notre Dame, Notre Dame, IN, United States of America, **4** Department of Electrical Engineering & Computer Science, University of Tennessee, Knoxville, TN, United States of America

* gwright3@nd.edu

**Data Availability Statement:** All relevant data are within the manuscript and its Supporting Information files.

**Funding:** This work was supported by National Institutes of Health (nih.gov) grant

## Abstract

Improved computational modeling of protein translation rates, including better prediction of where translational slowdowns along an mRNA sequence may occur, is critical for understanding co-translational folding. Because codons within a synonymous codon group are translated at different rates, many computational translation models rely on analyzing synonymous codons. Some models rely on *genome-wide* codon usage bias (CUB), believing that globally rare and common codons are the most informative of slow and fast translation, respectively. Others use the CUB observed only in *highly expressed* genes, which should be under selective pressure to be translated efficiently (and whose CUB may therefore be more indicative of translation rates). No prior work has analyzed these models for their ability to predict translational slowdowns. Here, we evaluate five models for their association with slowly translated positions as denoted by two independent ribosome footprint (RFP) count experiments from *S. cerevisiae*, because RFP data is often considered as a "ground truth" for translation rates across mRNA sequences. We show that all five considered models strongly associate with the RFP data and therefore have potential for estimating translational slowdowns. However, we also show that there is a weak correlation between RFP counts for the same genes originating from independent experiments, even when their experimental conditions are similar. This raises concerns about the efficacy of using current RFP experimental data for estimating translation rates and highlights a potential advantage of using computational models to understand translation rates instead.

## Introduction

A better understanding of the dynamics of protein translation (i.e., translation rates of ribosomes at specific codon positions along mRNA sequences) has many biological applications, such as enabling better understanding of co-translational protein folding and aiding in gene

1R01GM120733 awarded to SE, TM, PC, and JL. The funders had no role in study design, data collection and analysis, decision to publish, or preparation of the manuscript.

**Competing interests:** The authors have declared that no competing interests exist.

design for heterologous expression. Ribosome footprinting (RFP, also called ribosome profiling) is an experimental process often used to estimate ribosome tempo, i.e., the regional protein translation rate differences across a transcript [1, 2]. Briefly, cells are frozen, homogenized, and the ribosomes purified. The mRNA regions not covered by ribosomes are broken down with an enzyme, and the millions of remaining short fragments of ribosome-protected mRNA are then sequenced. The ribosome's A-site, where the amino acid-tRNA molecule binds to its corresponding codon, can then be estimated from alignments of these sequences to a reference; relatively higher estimates of ribosome occupancy suggest a slower rate of translation.

A well studied feature of non-uniform translation rates, and therefore of higher variability in RFP-inferred ribosome occupancy, is established codon preferences within most species ("codon usage bias", or CUB). Specifically, of the 20 standard amino acids, 18 have multiple codons that code for them. A group of codons that all code for the same amino acid are referred to as *synonymous* codons, and individual synonymous codons within genes have been shown to be translated at different rates [3–5]. Gene expression using mRNAs with only synonymous codon substitutions has been shown to alter protein folding mechanisms and the final protein structure formed [6–8]. A common notion in the literature is that "rare" codons (i.e., relatively infrequently used codons) are translated more slowly than other codons. There is debate, however, about when to consider a synonymous codon as rare and therefore slow. For example, some codon usage models such as %MinMax [9, 10] rely on genome-wide ("ORFeome") CUB. Other models have claimed that codon usage observed in highly expressed genes should be the best indicator of a given codon's translation speed, as highly expressed genes are likely under selective pressure for efficient translation [11, 12].

Codon preferences under these two categories of model bias can be drastically different. Take, for example, codons that code for the amino acid histidine ('GAC' and 'GAT'). Under ORFeome codon bias, 'GAC' is used to code for histidine about 65% of the time in *S. cerevisiae*. However, under the CAI model [12], which uses a pre-defined set of highly expressed genes to determine codon usage information, this preference is flipped—the codon 'GAT' is preferentially used about 64% of the time. This example of codon preference swapping based on gene expression level is just one of many examples of this phenomenon in *S. cerevisiae*.

Two recent studies have shown that rare codons (using different definitions for "rare") tend to occur at similar locations in orthologous genes found in a diverse collection of species ([13] used ORFeome CUB; [14] used CUB from highly expressed genes). These examples of rare codon co-occurrence imply a functional role for more slowly translated codons. To date, however, no prior effort has evaluated CUB models with respect to their ability to estimate locally slow translation. In this study we assess five different computational models for estimating translation rates (and therefore translational slowdowns) relative to experimental data (i.e., RFP data) that does the same. While each model is based on a distinct set of assumptions (see Methods), all result in a per-codon score where lower values imply slower translation than higher values. Conversely, in the experimental data, higher footprint counts at a given codon imply slower translation at said codon.

The contributions of this work are three-fold:

1. Because each considered computational model uses a sliding sequence-window to estimate translation tempo, we use a proof-of-concept classifier to confirm the window size that yields the most predictive power relative to RFP experimental data.

2. We evaluate how well each model's predicted slowly translated codon positions relate to experimental RFP data to determine which model is best associated with the data.

3. We compare RFP count distributions to quantify continuity between independent RFP experiments, and comment on the implications of our results.

## Methods

### Data processing

In this work we analyze five CUB models using two distinct RFP data sets from *S. cerevisiae*. The first data set, collected from [2] (NCBI GEO accession number GSE106572), contains already preprocessed mRNA reads mapped to their respective positions in the transcriptome. This accession contains RFP count data for 5,894 *S. cerevisiae* genes. Next, we map gene IDs from this prior study to a legacy *S. cerevisiae* ORFeome file containing 5,984 coding sequences, used to maintain consistency with other ongoing work (this file can be found in the *Supporting Information*). A sequence that matched in name and in length is assumed to be the same original sequence; this mapping removes six genes with no name match in our ORFeome file, and another 47 based on differing lengths. This data set is hereafter referred to as the Tunney data.

The second RFP data set is obtained from [15] (NCBI Sequence Read Archive SRR1049521). Unlike the first set, this data contains only the raw mRNA-Seq reads from [15]'s RFP experiment, which we downloaded as a FASTA file. Per [2], these reads are first pruned to remove any prefix of the ligated 3' linker TCGTATGCCGTCTTCTGCTTG from the end of the reads. Next, reads that align to ncRNA and rRNA are also removed [2]. The remaining reads are then aligned to the legacy ORFeome file using Bowtie2 [16], with options --norc (no reverse-compliment alignments), -a (all valid alignments were reported), and --gbar 30 (to prevent gapped alignments). These alignments are further pruned to remove alignments with more than 2 mismatches. Additionally, only reads of length 28-30nt are considered, as these allow for the most accurate assignment of A-sites per the supplement in [1]. For reads that map to a single position in the ORFeome, footprint counts are assigned per [1]'s supplement. FPKM values for each gene are determined using RSEM [17], which calculates estimated expression levels based on RNA seq alignments. Genes containing a multimapped read are assigned a footprint count equal to the FPKM for that gene divided by the sum total of FPKM values for all genes mapped to by said read. This data set is hereafter referred to as the Weinberg data.

The following processing steps are applied to all RFP data examined in this study. Because footprint counts at either end of a gene can be irregular, counts from the first 20 and last 20 codon positions of each gene are removed from consideration. Additionally, genes must have more than 200 net footprint counts (i.e., the sum total of the footprint counts in a gene must be greater than 200), and the number of positions with non-zero counts must be larger than 100. The data is then normalized on a per gene basis, with a gene's raw RFP counts divided by the average RFP count in that gene. These last four steps are inspired by [2] to help ensure data quality and comparability across genes, and remove a total of 1,779 sequences (30%) from consideration for the Tunney data and 1,173 sequences (19%) from the Weinberg data.

We also obtained 17 additional *S. cerevisiae* RFP data sets from 14 different studies available from GWIPS-vis [18]. Specific details of the 17 additional RFP data sets can be found in the *Supporting Information*. These data sets are chosen as a result of their high degree of similarity in library construction methods, *S. cerevisiae* strain used, and growth media. Nine of the 17 data sets contain biological replicates. After downloading, the files are converted from bigWig format to bedGraph format (which lists a position in the genome and respective footprint counts) using the bigWigToBedGraph binary utility available at the UCSC Genome Browser

([19], genome.ucsc.edu). The footprint counts are then mapped to their positions in annotated genes; genome files and annotations for the 2011 sacCer3 assembly are also available at the UCSC Genome Browser. This mapping creates a RFP count vector for each gene for each data set. These RFP count vectors are then subjected to the RFP filtering applied above.

In *RFP data sets are not very precise* we do a pairwise analysis of all 17 data sets. For each data set pair, a Pearson correlation coefficient is calculated for each RFP count vector of genes that appear in both data sets. These per-gene correlations are then averaged for each data set pair. Unlike the analysis done throughout the rest of this study, the analysis in this section does *not* map the RFP count vectors to the legacy ORFeome file, as coding sequence information is not needed, only the RFP counts themselves.

In the section *Specific codons appear to be "slow"* we examine how each of the different forms of codon usage bias (CUB *measures*) relates to RFP-implied slow codons from 14 GWIPS-vis data sets that use cycloheximide (CHX) to freeze the ribosomes (the remaining three data sets use a different method). Because this analysis relies on a mapping of RFP counts to individual codons, the post preprocessing RFP count vectors from the 14 data sets used are mapped to the legacy ORFeome sequences for consistency with the rest of the study. This extra mapping step removes no more than six sequences from any of the 14 data sets. Additionally, this section makes a distinction between CUB *measures* and CUB *models*. In short, CUB *measures* are different ways to quantify per-codon CUB preferences. These measures are then used as input into CUB *models* (that use sliding windows over sets of codons to make predictions along mRNA sequences about translation rates). There are only four CUB *measures* because both High-Phi %MinMax and High-Phi CAI (two of our five considered CUB *models*) are based on 'High-Phi' CUB *measurements*.

The models considered in this study (ORFeome %MinMax, High-Phi %MinMax, tAI, traditional CAI, and High-Phi CAI, defined in the Methods subsection Model Analysis), require a number of parameters as input. ORFeome codon usage frequencies for *S. cerevisiae* are obtained from HIVE-CUT [20]. CAI values are from [12]. tAI values are obtained from [21]. $\Delta\eta$ and $\Delta M$ values, necessary for calculating codon usage frequencies at varied expression levels per ROC-SEMPPR [11], are from Gilchrist (personal communication). To determine highly expressed codon usage frequencies ("High-Phi", for use in High-Phi %MinMax and High-Phi CAI) per [11], phi was set to 5.623.

## Window determination

All computational models analyzed in this work (outlined in *Model analysis*) utilize a sliding window over a set number of codons within mRNA sequences. %MinMax, and consequently the hybrid models we outlined in [22], have historically used a window size of 17 [13], with the A-site location being centered in the window. 17 was arbitrarily chosen as a compromise between smaller windows that were relatively noisy, and larger windows that could dilute an individual codon's contribution. Another study [2], which aims to predict RFP counts (and therefore local translation rates), found that a window of (-5, +4) around the A-site (i.e., 5 codons to the left of the A-site, the A-site codon itself, and four codons to the right of the A-site totalling 10 codons) was best correlated with empirical data in their neural network framework. This window size is in line with biological understanding of translational mechanisms, as the ribosome spans approximately 10 codons along an mRNA strand during translation [23].

We check whether another window size would be more appropriate for this analysis. Specifically, logistic regression (a common binary classification algorithm) is used to predict, using a

variety of input sliding window sizes, whether a given sequence position would have a RFP count either above or below a cutoff of:

1. The median RFP count in the data.

2. The average RFP count in the data.

3. The 90$^{th}$ percentile RFP count, defined as the RFP count resulting in the highest 10% of RFP counts belonging to a distinct class.

This process can be thought of as classifying positions as either translationally "fast" or "slow," using the above values as the cutoff between these two groups. Because of the 'exponential decay' shape of the RFP data, using the average RFP count as the cutoff for "slow" results in 32% and 37% of the data being labeled as such in the Tunney and Weinberg data, respectively. Intuitively, using the median and the 90th percentile RFP count results in 50% and 10% of the data labeled as slow. Each of the three versions of the classifier above are hereafter referred to as *instances* of the classifier.

Scikit-learn's logistic regression classifier [24] is used with an input feature of a one hot encoded vector in which each position contains the number of times a given codon occurs in the specified window. Each instance of the model takes in the one hot encoded vector and makes a prediction of 'slow' or 'not slow' for each codon position, based on the codons in the window around it. For each instance of the classifier, the data is randomly divided into five partitions such that each partition reflects the same ratio of "slow"-to-"fast" labels as the entire data set. The partitions are kept constant across all tested windows for a given classifier instance. Each classifier is trained and tested using 5-fold cross validation, and the classes are balanced during training to avoid overfitting. In our analysis: true positives are sequence positions that are predicted as slowly translated by the model and are labeled slowly translated by the RFP count data; true negatives are sequence positions that are both model-predicted and RFP count labeled as non-slowly translated; false positives are positions that are predicted to be slowly translated by the model and are not labeled slowly translated by the RFP count data; and false negatives are positions that are not predicted as slowly translated by the model, but are labelled slowly translated by the RFP count data. Tested window sizes vary from 1 to 21 for windows with the A-site positioned at the center. This range of window sizes is chosen to include a number slightly larger than the codon window historically used by %MinMax. When centering the A-site for even window sizes, the latter of the two possible middle positions is chosen to include the best predictive window from [2] (-5, +4) as an option in our analysis.

## Model analysis

In this work we analyze five computational models related to codon usage (ORFeome %Min-Max [9], High-Phi %MinMax [22], High-Phi CAI [22], traditional CAI [12], and tAI [25]) for their association with RFP-implied translational slowdowns. These models represent a number of different theories relating codon usage to translation rates in the literature. ORFeome % MinMax relies on genome-wide codon usage frequencies. tAI uses estimated tRNA levels to determine translationally fast and slow codons. We previously reported two hybrid expression bias models, High-Phi %MinMax and High-Phi CAI, based on ROC-SEMPPR high expression ("High-Phi") codon usage estimates [11]. We showed that both models correlate equally as well with empirically measured protein expression in *S. cerevisiae* [22] as traditional CAI, a model based on CUB in highly expressed genes which is also considered. While tAI and CAI have historically been used as global measures (i.e., one CAI or tAI value per gene), here we implement sliding windows to calculate a local per-codon score, based on the codons in the

sliding window around said codon. This allows for comparison with the other models and with the empirical RFP data.

One of our goals is to determine which of our analyzed models shows the strongest signal relative to RFP-implied translational slowdowns. To achieve this goal, we compare the distribution of RFP counts in two created bins (i.e., a predicted 'slow' bin and a predicted 'non-slow' bin) to determine whether the predicted slow bin contains higher overall RFP counts than the non-slow bin. Specifically, for each model, we analyze each gene in the cleaned data (see *Data processing*) by binning footprint counts based on whether a count's corresponding codon position is labeled slow or not by said model (a 'slow' prediction results from a position's model value being in the bottom 10% of all model values). The resulting two bins are then examined with a one-tailed Wilcoxon rank-sum test to determine if the count distribution in the slow bin is statistically significantly higher than in the non-slow bin. If statistical significance is found, this implies that the model predicted slowdowns are associated with translational slowdowns for the analyzed gene. To ensure enough data for the statistical test, both bins are required to contain at least 30 counts. Because our overall goal is to look for a data-wide association between experimental RFP counts and the models–and not to find individual genes that show significant differences in slow/not slow RFP count distributions–we rely on Fisher's method (also called Fisher's combined probability test) to aggregate the results of individual Wilcoxon rank-sum tests (one test per gene) and compute a single *p*-value per model/data set pair. To best balance data quantity with data quality (RFP data for 'denser' genes—genes that have a higher number of average footprint counts per position—are assumed to be less noisy), we run these tests on three groups of sequences:

1. All sequences that met the criteria set by [2] (see Data Processing) that also contain at least 30 counts in each bin.

2. The most dense 500 sequences in each RFP data set, as defined by the highest average RFP count per codon position, per [2]. This step should remove some noise found in RFP-count sparse genes. These sequences are then pruned to only include sequences that contain at least 30 counts in each bin.

3. The intersection of the sets of sequences used by each model in group 1, to allow for a fair comparison of *p*-values for each model. This set consists of 1,753 sequences for the Tunney data, and 1,914 sequences for the Weinberg data.

Additionally, to test whether any statistically significant signal found by Fisher's method is an artifact of our comparison framework, we repeat this analysis on group 3 after randomly shuffling the RFP counts for each gene 100 times per model, and report the average combined *p*-value for each model. Shuffling should decouple any relationships between individual codons and RFP-inferred occupancy and therefore is an appropriate null model for this analysis.

## Results

### Window determination

In agreement with the original analysis of [2], large increases in classifier performance are seen at window size 10 (from positions -5 to +4) across all instances of the classifier on the same data. Tunney *et al.* [2] also noted experimental artifacts in their RFP method that likely resulted in the codon positions -5 and +3 to be over-weighted. Precision, recall, and F1 for an alternative (-5, +3) window are plotted in Fig 1 as free standing points at window size 9 for the Tunney data. In all instances of the classifier, this new window outperforms the (-5, +4)

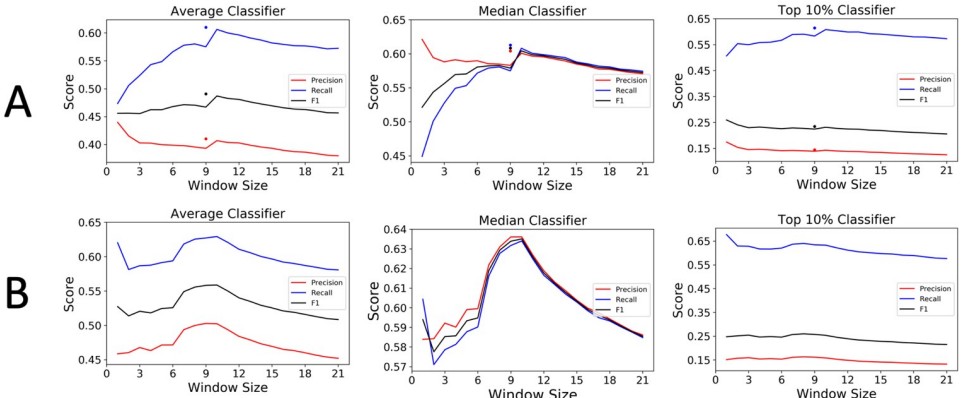

**Fig 1. Comparison of precision, recall, and F1 scores for different instances of the classifier.** For the Tunney data (A) clear jumps in performance are shown at window size 10 (from positions -5 to +4) for each classifier. Also shown are individual points for precision, recall and F1 score for the window (-5, +3). For the Weinberg data (B), the distinction between (-5, +3) values and (-5, +4) values is not as clear, although the window (-5, +4) does have the best F1 score on the Average and Median instances.

window on the Tunney data, but not on the Weinberg data. Peaks in F1 score for the Weinberg data are seen at window size 10 (-5, +4) in two of the three instances of the classifier. Because the window (-5, +3) is not as predictive as (-5, +4) on the Weinberg data, we provide further empirical support for the artifacts noted in [2]. Precision, recall, and F1 scores for each instance of the classifier are shown in Fig 1.

The use of empirical RFP data to parameterize the models is important since the distinction between the suggested windows and the traditional window of 17 (-8, +8) is sizeable. For example, the average Pearson correlation coefficient of ORFeome %MinMax values calculated for the windows (-5, +4) versus (-8, +8) for each gene is only 0.729, while the distinction between window size 9 (-5, +3) and window size 10 (-5, +4) is minor—the model values resulting from these windows have an average Pearson correlation coefficient of 0.943. For the remainder of this analysis we use the window size 10 (-5, +4) with each model.

## Model analysis

On the Tunney data, ORFeome %MinMax, tAI, and High-Phi CAI show a much stronger signal than High-Phi %MinMax and traditional CAI. However, the Weinberg data indicates a strong signal for all of the models. Distributions of individual *p*-values for All Sequence ORFeome %MinMax and Intersect CAI (the most and least statistically significant tests for the Tunney data, respectively) are shown in Fig 2. Results from Fisher's method, which aggregates the individual gene's *p*-values for each model, are shown in Table 1.

One notable result is that even the model that performs worst has a peak in its individual gene *p*-values on the left hand side of the graph in Fig 2, indicating a large number of individual genes that have a detectable, significant difference in translational tempo in line with model predictions. Additionally, the models are generally distinct in the genes that show a significant signal between predicted slow and non-slow positions, shown in Fig 3. That the various models are not finding strong associations between their predictions and the empirical data implies that they are different enough to warrant the analysis performed here.

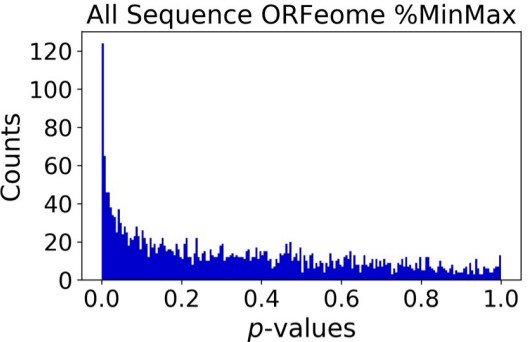
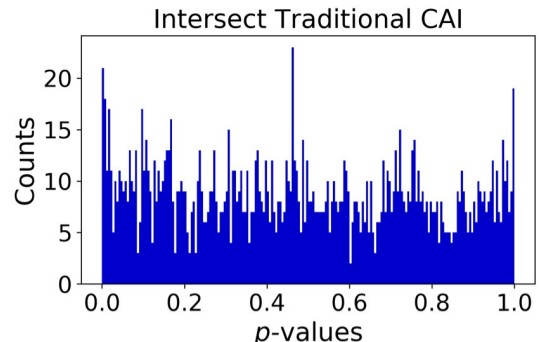

**Fig 2. Distribution of *p*-values for two representative tests on the Tunney data.** The test resulting in the most significant combined *p*-value (All Sequence ORFeome %MinMax, left) and the test resulting in the least significant combined p-value (Intersect Traditional CAI, right).

## RFP data sets are not very precise

Our underlying hypothesis is that codon usage not only has a significant association with slow translation–as shown above–but is also predictable and repeatable. It follows that, for a given gene in a given species, the translation tempo across the mRNA strand (represented by said gene's RFP count vector in the experimental data) would be highly correlated across different data sets. If this were the case, findings from studies that use RFP data to predict local translation rates (e.g., [1, 2]) would be largely independent of the data set used. However, we find that

**Table 1. The combined *p*-values and the number of sequences that passed filtering for each data partition and model pair (see Methods).** For the "Random" test, the reported *p*-value is the average *p*-value of 100 iterations of the null model described in the Methods. The "Intersect" partition is the intersection of the genes used for each model in "All Sequences".

| | | Tunney Data | | Weinberg Data | |
|---|---|---|---|---|---|
| **Test** | **Model** | **# of Sequences** | **Combined *p*-value** | **# of Sequences** | **Combined *p*-value** |
| All Sequences | ORFeome %MinMax | 2,614 | $p = 1 * 10^{-233}$ | 2,889 | $p < 2 * 10^{-308}$ |
| All Sequences | High-Phi %MinMax. | 2,462 | $p = 4 * 10^{-43}$ | 2,794 | $p < 2 * 10^{-308}$ |
| All Sequences | High-Phi CAI | 2,368 | $p = 5 * 10^{-88}$ | 2,707 | $p < 2 * 10^{-308}$ |
| All Sequences | Traditional CAI | 2,417 | $p = 3 * 10^{-25}$ | 2,745 | $p < 2 * 10^{-308}$ |
| All Sequences | tAI | 2,330 | $p = 6 * 10^{-153}$ | 2,671 | $p < 2 * 10^{-308}$ |
| Most Dense 500 | ORFeome %MinMax | 102 | $p = 6 * 10^{-30}$ | 101 | $p = 1 * 10^{-42}$ |
| Most Dense 500 | High-Phi %MinMax | 38 | $p = 2 * 10^{-34}$ | 12 | $p = 4 * 10^{-9}$ |
| Most Dense 500 | High-Phi CAI | 37 | $p = 4 * 10^{-78}$ | 7 | $p = 1 * 10^{-12}$ |
| Most Dense 500 | Traditional CAI | 36 | $p = 2 * 10^{-82}$ | 9 | $p = 2 * 10^{-13}$ |
| Most Dense 500 | tAI | 38 | $p = 2 * 10^{-57}$ | 10 | $p = 3 * 10^{-15}$ |
| Intersect | ORFeome %MinMax | 1,753 | $p = 2 * 10^{-128}$ | 1,914 | $p < 2 * 10^{-308}$ |
| Intersect | High-Phi %MinMax | 1,753 | $p = 4 * 10^{-23}$ | 1,914 | $p < 2 * 10^{-308}$ |
| Intersect | High-Phi CAI | 1,753 | $p = 6 * 10^{-50}$ | 1,914 | $p < 2 * 10^{-308}$ |
| Intersect | Traditional CAI | 1,753 | $p = 2 * 10^{-8}$ | 1,914 | $p < 2 * 10^{-308}$ |
| Intersect | tAI | 1,753 | $p = 4 * 10^{-90}$ | 1,914 | $p < 2 * 10^{-308}$ |
| Random | ORFeome %MinMax | 1,753 | $p = 0.983$ | 1,914 | $p = 0.732$ |
| Random | High-Phi %MinMax | 1,753 | $p = 0.970$ | 1,914 | $p = 0.770$ |
| Random | High-Phi CAI | 1,753 | $p = 0.986$ | 1,914 | $p = 0.757$ |
| Random | Traditional CAI | 1,753 | $p = 0.977$ | 1,914 | $p = 0.818$ |
| Random | tAI | 1,753 | $p = 0.985$ | 1,914 | $p = 0.780$ |

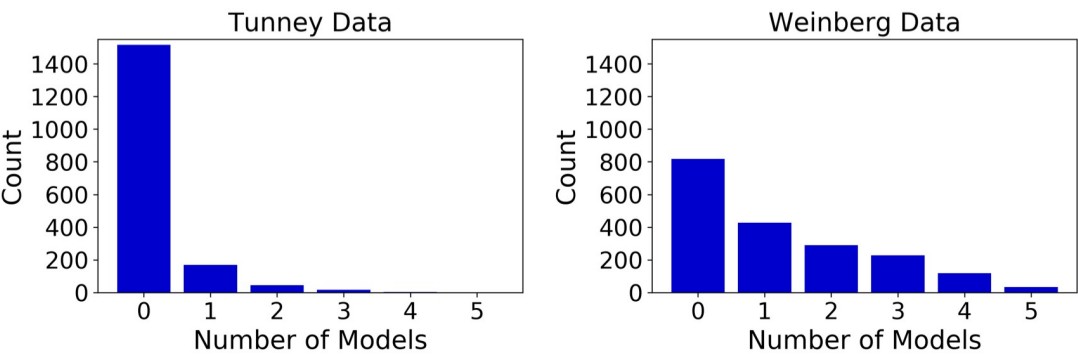

**Fig 3. Models are relatively distinct in the genes they determine have significant associations with RFP data.** In the Intersect partition, genes are grouped based on the number of models that predict each gene to have a significantly higher RFP count distribution in predicted slow positions than in predicted fast positions ($p < 0.01$).

RFP count vectors (post data preprocessing—see Methods) from genes assayed in independent studies are *not* generally well correlated (Fig 4), even if their experimental conditions are similar (Fig 5). The average Pearson correlation coefficient between the same gene in our two initial data sets (Tunney and Weinberg) is only 0.208 (Fig 4).

To further determine whether this problem is pervasive, we analyze 17 additional data sets downloaded from GWIPS-vis [18]. These 17 data sets are chosen because of the overall similarity of their experimental conditions (see Methods). 14 of the 17 use CHX to freeze the ribosomes during translation, while the other three data sets do not. Using the same comparison criteria as on the Tunney and Weinberg data, the pairwise correlations between each pair of the additional data sets (over all genes that appear in both data sets in the given pair) are shown in Fig 5. For the data sets that use CHX to freeze the ribosome (Fig 5A), the average Pearson correlation coefficient is only 0.1596, despite these data sets sharing similar experimental conditions. For the data sets that do not use CHX to freeze the ribosome (Fig 5B), the

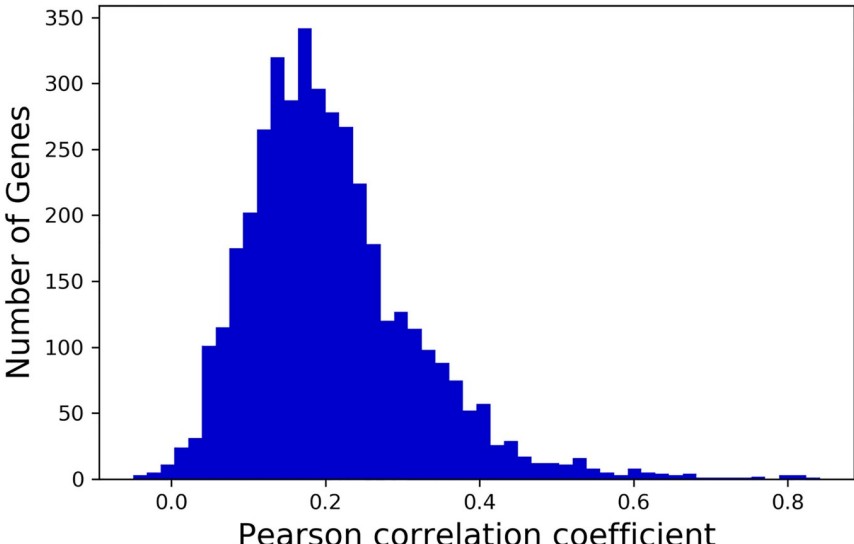

**Fig 4. Correlation of Weinberg and Tunney data sets.** Distribution of Pearson correlation coefficients for RFP count vectors of individual genes that appear in the Tunney and Weinberg data sets.

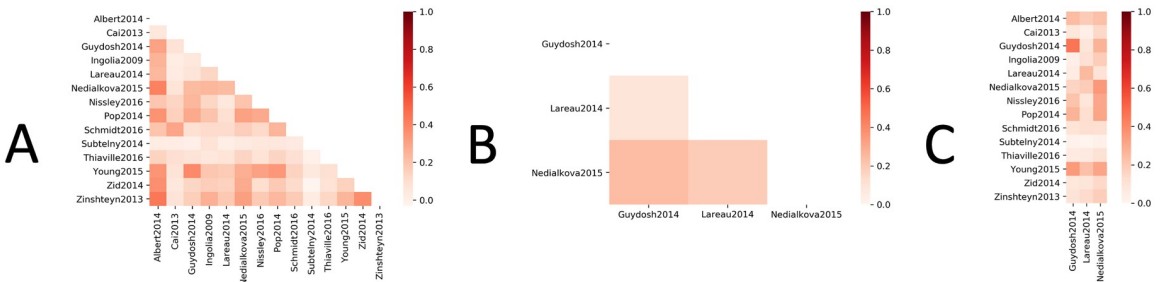

**Fig 5. Correlations of 17 RFP data sets.** (A) Average correlations between genes for data sets that use CHX to freeze the ribosome during translation. (B) Average correlations between genes for data sets that do not use CHX to freeze the ribosomes during translation. (C) Average correlation between genes between data sets that use CHX (y-axis) and data sets that do not (x-axis).

average Pearson correlation coefficient is 0.1769, despite these data sets sharing similar experimental conditions. The average Pearson correlation coefficient across these two groups, whose experimental conditions differ, is 0.1644 (Fig 5C). The distributions of correlations from the three groups of pairwise correlations depicted in Fig 5 show no statistically significant differences from each other (using a pair-wise Wilcoxon rank-sum test with significance threshold of 0.05). That is, the different RFP data sets correlate equally poorly no matter whether they have similar or dissimilar experimental conditions.

It should be noted that we are not the first group to observe a discrepancy between independent ribosome profiling experiments (i.e., between RFP data originating from different studies), although we are the first to compare RFP counts in the ORFeome across independent data sets; for further analysis on the subject see [26–28].

## Specific codons appear to be "slow"

One notable result from our Window Analysis is that the highest precision scores for all instances of the classifier occur when only the A-site is used (i.e., when the window size is one) on the Tunney data (Fig 1). Biochemically this makes sense—it is expected that the strongest influence on translation rate is the A-site codon, as the ribosome's A-site is where tRNA binding occurs.

To more comprehensively examine whether specific codons are enriched at the A-sites of high RFP count positions, we conducted a deeper, per-codon analysis of the 14 independent GWIPS-vis data sets that use CHX and have similar experimental conditions. For each of these 14 data sets, each non-stop codon's frequency in the top 10% of normalized RFP counts is compared to that same codon's frequency in the bottom 90% of footprint counts. These proportions are compared using Fisher's exact test with a $p$-value significance threshold of $8.2 * 10^{-4}$ (.05/61).

In total, 10 codons are found to be significantly over-enriched in high RFP count positions in at least 10 of the data sets (i.e., at least 70% of the 14 data sets analyzed), implying that these codons are generally translated more slowly. Additionally, another 13 codons are found to be significantly *under*-enriched in high RFP count positions in at least 10 of the data sets, implying that translational slowdowns generally do not occur at these codons. This large number of codons with significant frequency differences in high RFP count positions further suggests that individual codons have a substantial effect on translation, consistent with the belief that a ribosome's A-site should have the largest effect on translation tempo.

To determine how well codon enrichment in high RFP count positions align with each model's individual CUB measure, we assess each model's underlying CUB measure with

**Table 2. Codons that are significantly over-enriched in high RFP count positions in at least 10 of the 14 data sets considered (% Enriched > 70).** These codons are significantly enriched at the estimated A-site in the top 10% of normalized footprint counts using a Bonferroni corrected *p*-value of $8.2 * 10^{-4}$ (.05/61). These codons are also analyzed with respect to each bias measure, such that a larger negative number indicates a stronger correspondence with the model. Note that there are only four bias measures listed (compared to the five codon usage models analyzed earlier) as the High-Phi %MinMax and High-Phi CAI models use the same underlying CUB measure.

| Codon Information | | | Codon Usage Bias Measure | | | |
|---|---|---|---|---|---|---|
| Codon | AA | % Enriched | ORFeome | High-Phi | CAI | tAI |
| GGA | G | 0.857 | -0.024 | -0.240 | -0.248 | -0.146 |
| GAT | D | 0.857 | 0.151 | 0.020 | -0.144 | -0.195 |
| CCT | P | 0.857 | 0.060 | -0.052 | -0.206 | -0.130 |
| CCG | P | 0.857 | -0.126 | -0.245 | -0.248 | -0.058 |
| CCA | P | 0.857 | 0.157 | 0.522 | 0.695 | 0.351 |
| GAC | D | 0.786 | -0.151 | -0.020 | 0.144 | 0.195 |
| GGT | G | 0.714 | 0.205 | 0.678 | 0.725 | -0.008 |
| GGC | G | 0.714 | -0.053 | -0.197 | -0.230 | 0.302 |
| GAG | E | 0.714 | -0.201 | -0.386 | -0.484 | -0.184 |
| ACG | T | 0.714 | -0.112 | -0.238 | -0.247 | -0.160 |
| **Total** | | | **-0.093** | **-0.159** | **-0.244** | **-0.032** |

respect to the codons that are over- and under-enriched in high RFP count positions. Because "rare" codons are thought to be translated more slowly, a measure that has low codon usage frequencies for the over-enriched (i.e., RFP-implied slow) codons is likely a good predictor of individual codon's translation rates. Conversely, for under-enriched codons (i.e., RFP-implied faster codons), a measure that has high codon usage frequencies is likely a good predictor of codon translation rates. To test this, we subtract the frequency of each enriched codon under each CUB measure (i.e., ORFeome, High-Phi, traditional CAI, and tAI) with its expected frequency if synonymous codons were used at random (i.e., for a given codon, 1/(*numberofsynonyms*) in said codon's synonymous group). Note that High-Phi CUB is the same for both High-Phi %MinMax and High-Phi CAI—the underlying math is what differentiates the two models. This results in analyzing four *measures* of CUB, as opposed to the five different *models* that were analyzed in previous sections. For over-enriched codons in high RFP count positions, a CUB measure predicting these codons well (by having a small estimated frequency for each) will result in a larger negative number than a CUB measure performing less well. Conversely, for under-enriched codons, we would expect a better CUB measure to result in a larger, positive value. Results for each over-enriched codon, as well as the sum total for each CUB measure on over-enriched codons, can be seen in Table 2. While all CUB measures have some association with the enriched codons (due to the majority of codons under each model bias measure having a negative weight), the presence of a few very commonly used codons in each form of CUB prevent any of the CUB measures from differentiating themselves at aligning with RFP-implied slow codons. However, for significantly under-enriched codons, the high expression measures of CUB (CAI and High-Phi, scoring 2.412 and 1.963 respectively) outperformed tAI CUB (1.393) which in turn beat ORFeome CUB (0.317). This differentiation suggests that existing CUB measures are more adept at predicting which codons are likely to be translated efficiently, rather than which codons may cause translational slowdowns.

## Discussion

In this study, we were able to find broad consistencies across many independent *S. cerevisiae* RFP data sets, despite also finding an overall lack of correlation between count vectors of

individual genes across studies. This implies that some biological signal persists through the noise contained in RFP data sets.

First, when determining the codon window size to consider with our computational codon usage models, all instances of the classifier find that a window size between eight and 10 (specifically the windows (-4, +3), (-5, +3), and (-5, +4)) are the most predictive of RFP counts. While all of these windows are very similar to each other, they are very distinct from the window size traditionally used to study local translation rate—17 (-8, +8). Future uses of these sliding window codon usage models should rely on a smaller window than has historically been used; the window (-5, +4) is used in this analysis.

Using the window determined above, we proceed to examine five sliding window models for local translation rate to determine how well associated they are with RFP counts. Because of known effects of codon usage on overall protein folding in a cell [29], we are particularly interested in very low sliding window-based estimates (i.e., values in the bottom 10% for each model) and their association with high RFP counts, which imply slow translation at these positions. We calculate per-gene $p$-values that are then aggregated to determine the overall strength of signal between a model and RFP count data. Although the five models tested rely on different types of CUB measures, we find that all had statistically significant signal on both initial data sets. Interestingly, the three models that perform the best on the Tunney data (ORFeome %MinMax, High-Phi CAI, and tAI) are all based on different underlying assumptions of CUB. This supports the prior findings of [13, 14], who independently uncovered co-occurrence of rare codons (indicating potential functional roles for these codons) within orthologous proteins—one using ORFeome CUB and one using highly expressed CUB.

We next obtained a more comprehensive and comparable collection of RFP data consisting of a total of 17 data sets from highly similar yeast strains, growth media, and experimental conditions (see Methods). Through a pairwise analysis of these data sets, we show that RFP data between independent experiments are highly variable, even when experimental conditions are similar. Bioinformaticans, including ourselves, assume larger scale consistency versus the low actual correlations presented in Figs 4 and 5. However, these results suggest that translation tempo can differ across experiments, even when experimental conditions are kept largely constant. These results further emphasize the claims of previous studies [26–28] that improvement is needed in the ribosome profiling method.

Finally, using a subset of the additional 17 data sets that were the most experimentally similar (14 total), we identify 10 codons that are significantly enriched in the top 10% of RFP counts in at least 70% of the analyzed data sets (Table 2). An additional 13 codons are found to be significantly *under* enriched in the top 10% of normalized RFP counts. These findings support the strong effect that A-site codon usage has on translation—both slow and non-slow. We analyze how these significantly over- and under-enriched codons compare to CUB measures tested in this work. While all of the measures considered show some degree of correspondence, in agreement with earlier studies ([13, 14]) and our Fischer's tests shown in Table 1, no form of CUB stood out when predicting translationally slow codons. However, the same test on significantly under-enriched codons showed a much stronger correspondence with CUB from highly expressed genes (i.e., "High-Phi" and "CAI").

In Table 2, one of the codons with the worst (most positive) scores across all models is 'CCA', which is the most abundant proline (amino acid 'P') codon across all models. In fact, proline is known to substantially slow translation [30, 31]. This observation is consistent with our results (as three of the four proline codons are over-enriched in high RFP count positions in both data sets) and suggests that amino acid effects on translation can overshadow individual codon effects.

## Conclusion

Local translation rate is thought to have significant effects on co-translational folding. One established proxy for local translation rate is biased codon usage, where *rare* codons are assumed to be translated more slowly than *common* codons. There is some debate, however, about how to define rare and common. ORFeome CUB defines rare and common codons by usage rates across all predicted genes for a given organism. On the other hand, highly expressed CUB defines rare and common codons by usage rates only in highly expressed genes. Both forms of CUB have support in the literature. Here we test five different computational codon usage models (in which both types of CUB are represented) for their association with high RFP count positions in *S. cerevisiae*. Because the computational models tested in this study rely on sequence sliding windows, we also use a proof-of-concept classifier to determine which window is most predictive of RFP count data, and therefore a better proxy for translation tempo.

We independently confirm that a sequence window of positions (-5, +4) around the A-site is most predictive of translation rate. We also show that computational codon usage models from all three tested forms of underlying CUB (ORFeome, highly expressed, and predicted tRNA concentration) are globally associated with experimentally inferred translational slowdowns (i.e., high RFP count positions). Additionally, we show that 10 codons are significantly over-enriched in these high RFP count positions, implying that they are more slowly translated than other codons. We also found 13 codons that are significantly under-enriched in very high RFP count positions, and that these codons are better associated with models implementing CUB from highly expressed genes than genome-wide measures. All models discussed in this work have been incorporated into a novel experimental approach (based on [10]) to determine which model performs best at estimating local translation rates *in vivo*.

Finally, we also support prior concerns about the ability of ribosome footprinting to define translation rates at a per-codon resolution. Future work in this area, which can be aided by our computational approach, is needed to determine the biological factors that affect translation tempo.

## Supporting information

**S1 Table.**
(DOCX)

**S2 Table.**
(CSV)

## Author Contributions

**Conceptualization:** Gabriel Wright, Scott J. Emrich.

**Data curation:** Gabriel Wright, Anabel Rodriguez.

**Formal analysis:** Gabriel Wright.

**Funding acquisition:** Jun Li, Patricia L. Clark, Tijana Milenković, Scott J. Emrich.

**Investigation:** Gabriel Wright.

**Methodology:** Gabriel Wright.

**Software:** Gabriel Wright.

**Supervision:** Tijana Milenković, Scott J. Emrich.

**Visualization:** Gabriel Wright.

**Writing – original draft:** Gabriel Wright, Scott J. Emrich.

**Writing – review & editing:** Gabriel Wright, Anabel Rodriguez, Jun Li, Patricia L. Clark, Tijana Milenković, Scott J. Emrich.

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
