## [Decision Letter · Decision Letter 0]

24 Jan 2020

PONE-D-19-31094

Analysis of computational codon usage models and their association with translationally slow codons

PLOS ONE

Dear Mr. Wright,

Thank you for submitting your manuscript to PLOS ONE. After careful consideration, we feel that it has merit but does not fully meet PLOS ONE’s publication criteria as it currently stands. Therefore, we invite you to submit a revised version of the manuscript that addresses the points raised during the review process.

We would appreciate receiving your revised manuscript by Mar 09 2020 11:59PM. To enhance the reproducibility of your results, we recommend that if applicable you deposit your laboratory protocols in protocols.io, where a protocol can be assigned its own identifier (DOI) such that it can be cited independently in the future. For instructions see: http://journals.plos.org/plosone/s/submission-guidelines#loc-laboratory-protocols

We look forward to receiving your revised manuscript.

Kind regards,

Zhi Xie

Academic Editor

PLOS ONE

Reviewers' comments:

Reviewer's Responses to Questions

**Comments to the Author**

1. Is the manuscript technically sound, and do the data support the conclusions?

Reviewer #1: No

Reviewer #2: Yes

2. Has the statistical analysis been performed appropriately and rigorously? 

Reviewer #1: Yes

Reviewer #2: Yes

3. Have the authors made all data underlying the findings in their manuscript fully available?

Reviewer #1: Yes

Reviewer #2: Yes

4. Is the manuscript presented in an intelligible fashion and written in standard English?

Reviewer #1: Yes

Reviewer #2: Yes

5. Review Comments to the Author

Reviewer #1: In this manuscript, authors assessed five different computational models for estimating translation rates relative to experimental data that does the same. It's interesting but some improvements/modifications are required before receiving the publication.

1) In line 78 on page 3, I think you mean "more than" not "more then"?

2) Table 2, why are the results for only four models displayed?

3) Figure 4A, I searched the two RFP datasets at NCBI, and they didn't have biological replicates. As far as I know, the Pearson correlation coefficient of a gene in two studies was not statistically significant.

4) Figure 4B, are the experimental conditions of these 26 data sets consistent? For example, library construction methods, experimental process of yeast, etc.

Reviewer #2: The manuscript written by Wright et al. addressed an important question in the field. Overall, the manuscript was well written. However, I have three concerns, particularly the first and the second. The authors should address the concerns.

Major concerns

1. For the section “Specific codons appear to be slow”, the statement was based on two datasets. It’s not clear whether this statement was specific to these two datasets or general to different data. I therefore would suggest the authors conduct this analysis on more datasets.

2. For the section “RFP datasets are not very precise”, the authors analyzed 26 independent RFP datasets. The authors should be more specific on the datasets. Whether these datasets include biological replicates or what conditions are the experiments? Some correlation is 0.79 while average correlation is 0.199. Clearly the translation speed is related to specific conditions but the authors completely ignored it.

3. Figure 1: it is not clear to me how the positive and negative instances were defined when calculating precision, recall and F1 scores.

6. PLOS authors have the option to publish the peer review history of their article (what does this mean?). If published, this will include your full peer review and any attached files.

Reviewer #1: No

Reviewer #2: No

---

## [Author Response · Author response to Decision Letter 0]

11 Mar 2020

Reviewer 1:

In this manuscript, authors assessed five different computational models for estimating translation rates relative to experimental data that does the same. It's interesting but some improvements/modifications are required before receiving the publication.

1) In line 78 on page 3, I think you mean "more than" not "more then"?

Fixed. 

2) Table 2, why are the results for only four models displayed?

In the paper we make a distinction between codon usage bias (CUB) models and CUB measures. In short, CUB models are used as translation rate predictors along mRNA sequences using (i) sliding windows, (ii) input per-codon usage scores, and (iii) an underlying mathematical formula. CUB measures are the per-codon scores that are used as input into the CUB models. In our paper, we analyze five different CUB models. Because two of these models (“High-Phi %MinMax” and “High-Phi CAI”) use the same CUB measure (“High-Phi”), our study has four CUB measures instead of five. The analysis in the section Specific codons appear to be “slow” (containing Table 2) focuses on individual codons, and not codons aggregated over sliding windows. Therefore this section analyzes the results with respect to the four CUB measures, not five CUB models. To make this clear, we have added details in a paragraph of the Methods subsection Data processing, beginning at line 124. We also added additional clarification in the section Specific codons appear to be “slow” in lines 347-348, and in the Table 2 caption.

3) Figure 4A, I searched the two RFP datasets at NCBI, and they didn't have biological replicates. As far as I know, the Pearson correlation coefficient of a gene in two studies was not statistically significant.

First, regarding the correlation coefficient: the lack of correlation between the different RFP data sets is exactly what the conclusion of our analysis is. The Tunney paper that one of the sets is pulled from proposed a neural network model of translation and claimed that this model “captures information determining translation dynamics in vivo”. However, the fact that we observe a lack of correlation between the data sets means that the above model (or any of the many models that have been built from RFP data), when trained on one data set, would not be an accurate predictor of translation dynamics when applied to a different data set. 

Second, regarding the replicates: It is true that the two data sets in Figure 4A do not have biological replicates. Of the 17 RFP data sets that we have analyzed from GWIPS-vis (i.e., the data sets that are analyzed in the newly added Figure 5), nine have replicates. Moreover, as the new Figure 5 shows, even when we control for experimental protocols/conditions under which the different RFP data sets were derived (see comment 4 below for details), we find that regardless of whether the data sets have replicates or not, correlations between all datasets are still low. Namely, even the highest Pearson’s correlation coefficient is below 0.5, and the average Pearson correlation coefficient over all considered data set pairs is below 0.2. Therefore our findings hold irrespective of the number of replicates.

4) Figure 4B, are the experimental conditions of these 26 data sets consistent? For example, library construction methods, experimental process of yeast, etc.

This is a fair question, and we have since revised our study to hopefully provide a more acceptable answer. From the 26 studies originally analyzed, we now choose 14 that are largely similar in library construction method, yeast strain, and growth media. From these 14 studies we now analyze 17 data sets, nine of which have biological replicates. Of these 17 data sets, first, we choose only the data sets of wild-type yeast where the ribosomes are frozen using cycloheximide; we compare (i.e., correlate) these 14 same-experiment data sets in Figure 5A. Second, from the same 17 data sets, we separately choose three data sets of wild-type yeast where the ribosomes were frozen with a method that does not use cycloheximide; we compare (i.e., correlate) these three same-experiment datasets in Figure 5B. Third, we compare (i.e., correlate) the data sets across the above two experimental conditions in Figure 5C. Interestingly, after controlling for experimental conditions, we find that our results are qualitatively the same as before: we find a stark lack of correlation between the data sets in all three cases (regardless of whether we control for the method used to freeze the ribosome or not). These results further emphasize our claim that RFP data sets should be used with caution, as they are not ideal ground truths of translation rates. 

Reviewer 2:

The manuscript written by Wright et al. addressed an important question in the field. Overall, the manuscript was well written. However, I have three concerns, particularly the first and the second. The authors should address the concerns.

Major concerns

1. For the section “Specific codons appear to be slow”, the statement was based on two datasets. It’s not clear whether this statement was specific to these two datasets or general to different data. I therefore would suggest the authors conduct this analysis on more datasets.

We have heeded the reviewer’s concern and now run this analysis on additional data sets. Specifically, we now run this analysis on the 14 RFP data sets that have similar experimental conditions and use cycloheximide as a ribosome freezing agent (see response to comment 4 of Reviewer 1). These new results can be found in the updated Table 2, as well as throughout the section Specific codons appear to be “slow”. Briefly, the results are qualitatively the same. While no codons are enriched across all 14 data sets, we find 10 that are statistically significantly enriched in high RFP count positions in at least 70% of the data sets (10/14). These 10 codons include codons that code for amino acids that are known to be slowly translated (such as three Proline codons). 

2. For the section “RFP datasets are not very precise”, the authors analyzed 26 independent RFP datasets. The authors should be more specific on the datasets. Whether these datasets include biological replicates or what conditions are the experiments? Some correlation is 0.79 while average correlation is 0.199. Clearly the translation speed is related to specific conditions but the authors completely ignored it.

Done, as already explained in our response to comment 4 of Reviewer 1. Brief descriptions of all data sets used can be found in the Supporting Information, as well as directions for navigating GWIPS-vis to access all of the information that they have made available.

3. Figure 1: it is not clear to me how the positive and negative instances were defined when calculating precision, recall and F1 scores.

Explicit definitions of true positives, true negatives, false positives, and false negatives have been added in Methods subsection Window determination, beginning at line 177.

---

## [Decision Letter · Decision Letter 1]

7 Apr 2020

Analysis of computational codon usage models and their association with translationally slow codons

PONE-D-19-31094R1

Dear Dr. Wright,

We are pleased to inform you that your manuscript has been judged scientifically suitable for publication and will be formally accepted for publication once it complies with all outstanding technical requirements.

With kind regards,

Zhi Xie

Academic Editor

PLOS ONE

Additional Editor Comments (optional):

Reviewers' comments:

Reviewer's Responses to Questions

**Comments to the Author**

1. If the authors have adequately addressed your comments raised in a previous round of review and you feel that this manuscript is now acceptable for publication, you may indicate that here to bypass the “Comments to the Author” section, enter your conflict of interest statement in the “Confidential to Editor” section, and submit your "Accept" recommendation.

Reviewer #1: (No Response)

Reviewer #2: All comments have been addressed

2. Is the manuscript technically sound, and do the data support the conclusions?

Reviewer #1: (No Response)

Reviewer #2: Yes

3. Has the statistical analysis been performed appropriately and rigorously? 

Reviewer #1: (No Response)

Reviewer #2: Yes

4. Have the authors made all data underlying the findings in their manuscript fully available?

Reviewer #1: (No Response)

Reviewer #2: Yes

5. Is the manuscript presented in an intelligible fashion and written in standard English?

Reviewer #1: (No Response)

Reviewer #2: Yes

6. Review Comments to the Author

Reviewer #1: (No Response)

Reviewer #2: (No Response)

7. PLOS authors have the option to publish the peer review history of their article (what does this mean?). If published, this will include your full peer review and any attached files.

Reviewer #1: No

Reviewer #2: No

---

## [Editor Report · Acceptance letter]

17 Apr 2020

PONE-D-19-31094R1 

Analysis of computational codon usage models and their association with translationally slow codons 

Dear Dr. Wright:

I am pleased to inform you that your manuscript has been deemed suitable for publication in PLOS ONE. Congratulations! Your manuscript is now with our production department. 

With kind regards,

on behalf of

Dr. Zhi Xie 

Academic Editor

PLOS ONE